# VRKitchen: an Interactive 3D Environment for Learning Real Life Cooking Tasks

**Xiaofeng Gao** [1]  **Ran Gong** [1]  **Tianmin Shu** [1]  **Xu Xie** [1]  **Shu Wang** [1]  **Song-Chun Zhu** [1]

## Abstract

One of the main challenges of applying reinforcement learning to real world applications is the lack of realistic and standardized environments for training and testing AI agents. In this work, we design and implement a virtual reality (VR) system, VRKitchen, with integrated functions which i) enable embodied agents to perform real life cooking tasks involving a wide range of object manipulations and state changes, and ii) allow human teachers to provide demonstrations for training agents. We also provide standardized evaluation benchmarks and data collection tools to facilitate a broad use in research on learning real life tasks. Video demos, code, and data will be available on the project website: `sites.google.com/view/vr-kitchen/`.

## 1. Introduction

Thanks to the recent success in many domains of AI research, humans now have built machines that can accurately detect and recognize objects (Krizhevsky & Hinton, 2012; He et al., 2017), generate vivid natural images (Brock et al., 2018), and beat human Go champions (Silver et al., 2017). However, a truly intelligent machine agent should be able to solve a large set of complex tasks in the real world, by adapting itself to unseen surroundings and planning a long sequence of actions to reach the desired goals, which is still beyond the capacity of current machine models. This gives rise to the need of advancing research on learning real life tasks. For the present work, we are interested in the following three aspects of real life task learning problem.

**Learning visual representation of a dynamic environment**. In the process of solving a task in a dynamic environment, the appearance of the same object may change dramatically as a result of actions (Isola et al., 2015; Fathi

& Rehg, 2013; Liu et al., 2017b). To capture such variation in object appearance, the agent is required to have a better visual representation of the environment dynamics. To make a dish with a tomato, for example, the agent should recognize the tomato even if it is cut into pieces and put into container. To acquire such visual knowledge, it is important for an agent to learn from physical interactions and reason over the underlying causality of object state changes. There have been work on implementing interaction-based learning in lab environments (Lerer et al., 2016; Agrawal et al., 2015; Haidu et al., 2015), but the limited scenarios greatly restrict scalability and reproducibility of prior work. Instead, we believe that building a simulation platform is a good alternative since i) performance of different algorithms can be easily evaluated and benchmarked, and ii) a large set of diverse and realistic environments and tasks can be customized.

**Learning to generate long-term plans for complex tasks.** In real life scenarios, a complex task is often composed of various sub-tasks, each of which has its own sub-goal (BARNES, 1944). Thus the agent needs to take a long sequence of actions to finish the task. The large number of possible actions in the sample space and the extremely sparse rewards make it difficult to steer the policy to the right direction. Recently, many researchers have focused on learning hierarchical policies (Stolle & Precup, 2002; Andreas et al., 2016; Shu et al., 2018) in simple domains. In this work, we provide a virtual environment where the agent can learn to compose long-term plans for daily life tasks that humans encounter in the real world.

**Learning from human demonstrations to bootstrap agents' models.** Training an agent from scratch is extremely difficult in complex environments. To bootstrap the training, it is common to let an agent to imitate human experts by watching human demonstrations (Ng & Russell, 2000; Ziebart et al., 2008; Giusti et al., 2016). Previous work has shown that learning from demonstrations (or imitation learning) significantly improves the learning efficiency and achieves a higher performance than reinforcement learning does (Zhu et al., 2017; Hester et al., 2017). However, it is expensive and time consuming to collect diverse human demonstrations with high qualities. We believe that virtual reality platforms can provide us with an ideal medium to crowd source demonstrations from a broad range of users

---

[1] Center for Vision, Cognition, Learning and Autonomy, University of California, Los Angeles, USA. Correspondence to: Xiaofeng Gao <xfgao@ucla.edu>.

*Reinforcement Learning for Real Life (RL4RealLife) Workshop in the 36th International Conference on Machine Learning*, Long Beach, California, USA, 2019. Copyright 2019 by the author(s).

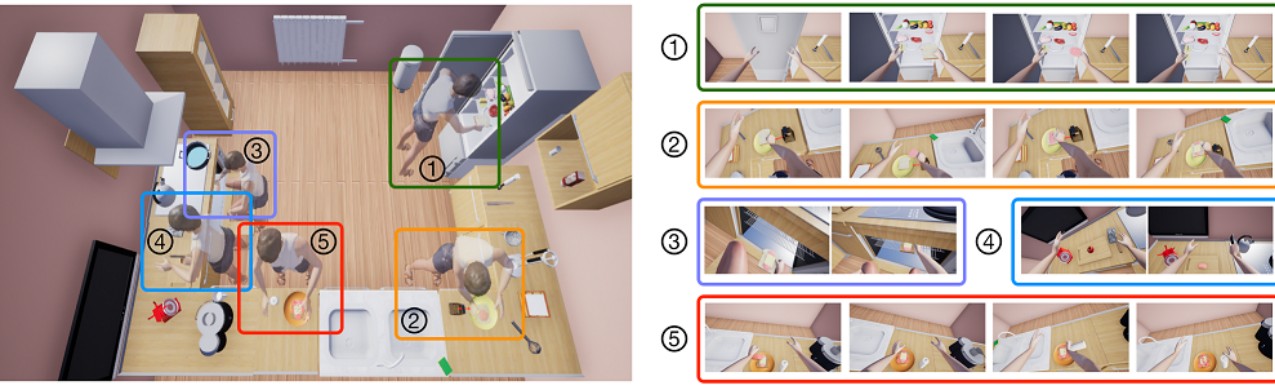

Figure 1. A sample sequence of an agent making a *sandwich*. Rectangles on the left graph represents five necessary sub-tasks, including (1) taking ingredients from fridge, (2) putting ham and cheese on the bread, (3) use the oven, (4) cut tomato and (5) add some sauce. Each rectangle on the right graph indicates atomic actions required to finish a sub-task.

(von Ahn & Dabbish, 2008).

In this work, we focus on simulating two sets of real life cooking tasks (using common tools and preparing dishes) in a virtual kitchen environment, VRKitchen. We illustrate how this system can address the emerged needs for real life task learning in an example shown in Figure 1, where an agent makes a sandwich in one of the kitchens created in our system.

1. Cooking tasks contain a notoriously great number of changes in object states. VRKitchen allows the agent to interact with different *tools* and *ingredients* and simulates a variety of object state changes. E.g., the bread changes its color when it is being heated in the oven, and the tomato turns into slices after it is cut. The agent's interactions with the physical world when performing cooking tasks will result in large variations and temporal changes in objects' appearance and physical properties, which calls for a task-oriented visual representation.

2. To make a sandwich, the agent needs to perform a long sequence of actions, including taking ingredients from a fridge, putting cheese and ham on the bread, toasting the bread, adding some sliced tomato and putting some sauce on the bread. To quickly and successfully reach the final goal, it is necessary to equip the agent with the ability to conduct long-term planning.

3. We build two interfaces to allow an AI algorithm as well as a human user to control the embodied agent respectively, thus humans can give demonstrations using VR devices, and the AI algorithms can learn from these demonstrations and perform the same tasks in the same virtual environments.

In summary, our main contributions are:

- A 3D virtual kitchen environment which enables physical simulation of a wide range of cooking tasks with rich object state changes and compositional goals;
- A toolkit including a VR-based user interface for collecting human demonstrations, and a Python API for training

and testing different AI algorithms in the virtual environments.

- Proposing a new challenge – VR chef challenge, to provide standardized evaluation for benchmarking different approaches in terms of their learning efficiency in complex 3D environments.

- A new human demonstration dataset of various cooking tasks – UCLA VR chef dataset.

## 2. Related Work

**Simulation platforms.** Traditionally, visual representations are learned from static datasets. Either containing prerecorded videos (Rohrbach et al., 2012) or images (Jia Deng et al., 2009), most of them fail to capture the dynamics in viewpoint and object state during human activities, in spite of their large scale.

To address this issue, there has been a growing trend to develop 3D virtual platforms for training embodied agents in dynamic environments. Typical systems include 3D game environments (Kempka et al., 2017; Beattie et al., 2016; Johnson et al., 2016), and robot control platforms (Todorov et al., 2012; Coumans & Bai, 2016; Fan et al., 2018; Plappert et al., 2018). While these systems offer physics simulation and 3D rendering, they fail to provide realistic environments and daily tasks humans face in the real world.

More recently, based on 3D scene datasets such as Matterport3D (Chang et al., 2018) and SUNCG (Song et al., 2017), there have been several systems simulating more realistic indoor environments (Brodeur et al., 2017; Wu et al., 2018; Savva et al., 2017; McCormac et al., 2017; Xia et al., 2018; Xie et al., 2019) for visual navigation tasks and basic object interactions such as pushing and moving funitures (Kolve et al., 2017). While the environments in these systems are indeed more realistic and scalable compared to previous systems, they can not simulate complex object manipulation that are common in our daily life. (Puig et al., 2018) took a

| Env. | Large-scale | Physics | Realistic | State | Manipulation | Avatar | Demo |
|---|---|---|---|---|---|---|---|
| Malmo (Johnson et al., 2016) | √ | √ | | √ | | | |
| DeepMind Lab (Beattie et al., 2016) | | √ | | | | | |
| VizDoom (Kempka et al., 2017) | | √ | | | | | |
| MINOS (Savva et al., 2017) | √ | | √ | | | | |
| HoME (Brodeur et al., 2017) | √ | √ | √ | | | | |
| Gibson (Xia et al., 2018) | √ | √ | √ | | | √ | |
| House3D (Wu et al., 2018) | √ | √ | √ | | | | |
| AI2-THOR (Kolve et al., 2017) | | √ | √ | √ | | | |
| VirtualHome (Puig et al., 2018) | | | √ | √ | √ | √ | |
| SURREAL (Fan et al., 2018) | | √ | | | √ | | √ |
| VRKitchen (ours) | | √ | √ | √ | √ | √ | √ |

*Table 1.* Comparison with other 3D virtual environments. Large-scale: a large number of scenes. Physics: physics simulation. Realistic: photo-realistic rendering. State: changeable object states. Manipulation: enabling fine-grained object interactions and manipulations. Avatar: humanoid virtual agents. Demo: user interface to collect human demonstrations.

step forward and has created a dataset of common household activities with a larger set of agent actions including pick-up, switch on/off, sit and stand-up. However, this system was designed to generate data for video understanding. In contrast, our system emphasizes training and evaluating agents on real life cooking tasks, which involves fine-grained object manipulation on the level of object parts (e.g., grasping the handle of a knife), and flexible interfaces for allowing both human users and AI algorithms to perform tasks. Our system also simulates the animation of object state changes (such as the process of cutting a fruit) and the gestures of humanoid avatars (such as reaching for an object) instead of only showing pre-conditions and post-effects as in (Kolve et al., 2017). A detailed comparison between our system and other virtual environments is summarized in Table 1.

**Imitation learning.** Learning from demonstration or imitation learning is proven to be an effective approach to train machine agents efficiently (Abbeel & Ng, 2004; Syed & Schapire, 2008; Ross et al., 2010). Collecting diverse expert demonstrations with 3D ground-truth information in real world is extremely difficult. We believe the VR interface in our system can greatly simplify and scale up the demonstration collection.

**VR for AI.** VR provides a convenient way to evaluate AI algorithms in tasks where interaction or human involvement is necessary. Researches have been conducted on relevant domains, including physical intuition learning (Lerer et al., 2016), human-robot interaction (Shu et al., 2016; 2017; Liu et al., 2017a; de Giorgio et al., 2017), learning motor control from human demonstrations (Liu et al., 2019; Haidu et al., 2015; Kawasaki et al., 2001; Belousov et al., 2001). Researchers have also used VR to collect data and train computer vision models (Zhong et al., 2019). To this end, several plugins for game engines have been released, such as UETorch (Lerer et al., 2016) and UnrealCV (Qiu & Yuille, 2016). To date, such plugins only offer APIs to control game state and record data, requiring additional packages to train virtual agents.

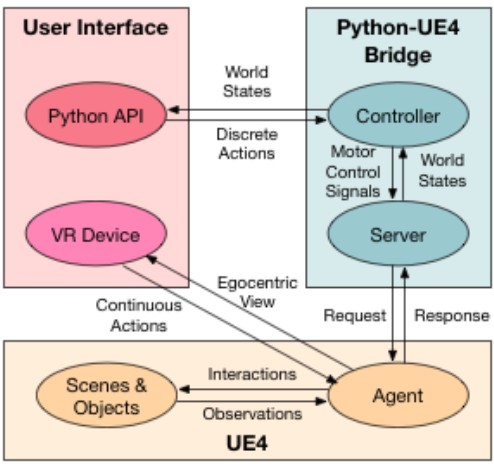

*Figure 2.* Architecture of VRKitchen. Users can either directly teleoperate the agent using VR device or send commands to the agent by Python API.

## 3. VRKitchen Environment

Our goal is to enable better learning of autonomous agents for real life tasks with compositional goals and rich object state changes. To this end, we have designed VRKitchen, an interactive virtual kitchen environment which provides a testbed for training and evaluating various learning and planning algorithms in a variety of cooking tasks. With the help of virtual reality device, human users serve as teachers for the agents by providing demonstrations in the virtual environment.

### 3.1. Architecture Overview

Figure 2 gives an overview of the architecture of VRKitchen. In particular, our system consists of three modules: (1) the physics engine and photo-realistic rendering module consists of several humanoid agents and kitchen scenes, each

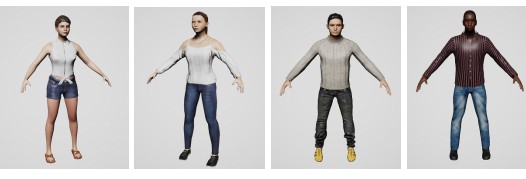

(a) Female 1  (b) Female 2  (c) Male 1  (d) Male 2

*Figure 3.* Four humanoid avatars designed using MakeHuman (The MakeHuman team, 2017).

has a number of ingredients and tools necessary for performing cooking activities; (2) a user interface module which allows users or algorithms to perform tasks by virtual reality device or Python API; (3) a Python-UE4 bridge, which transfers high level commands to motor control signals and sends them to the agent.

### 3.2. Physics Engine and Photo-realistic Rendering

As a popular game engine, Unreal Engine 4 (UE4) provides physics simulation and photo-realistic rendering which are vital for creating a realistic environment. On top of that, we design humanoid agents, scenes, object state changes, and fine-grained actions as follows.

**Humanoid agents**. Agents in VRKitchen have human-like appearances (shown in Figure 3) and detailed embodiment representations. The animation of the agent can be broken into different states, e.g. *walking, idle*. Each agent is surrounded by a capsule for collision detection: when it's *walking*, it would fail to navigate to a new location if it collides with any objects in the scene. When it is *idle*, the agent can freely interact with objects within certain range of its body.

**Scenes**. VRKitchen consists of 16 fully interactive kitchen scenes as shown in Figure 4. Agents can interact with most of the objects in the scenes, including various kinds of *tools, receptacles* and *ingredients*. Each kitchen is designed and created manually based on common household setting. 3D models of furnitures and appliances in kitchens are first obtained from the SUNCG dataset (Song et al., 2017). Some of the models are decomposed to create necessary object interactions, e.g. we reassemble doors and cabinets to create effects for opening and closing the door. After we have basic furnitures and appliances in the scene, we then add cooking *ingredients* and *tools*. Instead of sampling their locations randomly, we place the objects according to their utility, e.g. *tools* are placed on the cabinets while perishable *ingredients* such as fruits and vegetables are available in the fridge. On average, there are 55 interactive objects in a scene.

**Object state changes**. One key factor of VRKitchen is the ability to simulate state changes for objects. Instead of showing only pre-conditions and post effects of actions, VRKitchen simulates the continuous geometric and topological changes of objects caused by actions. This leads

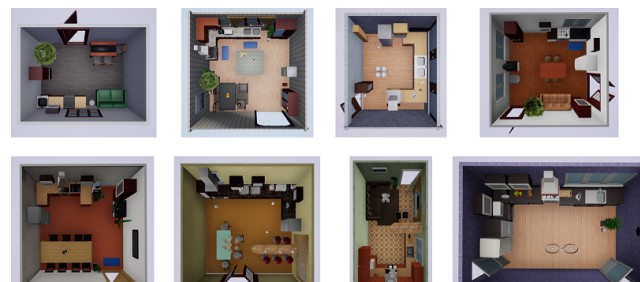

*Figure 4.* Sample kitchen scenes available in VRKitchen. Scenes have a variety of appearance and layouts.

to a great number of available cooking activities, such as roasting, peeling, scooping, pouring, blending, juicing, etc. Overall, there are 18 cooking activities available in VRKitchen. Figure 5 shows some examples of object interactions and state changes.

**Fine-grained actions**. In previous platforms (Kolve et al., 2017; Brodeur et al., 2017), objects are typically treated as a whole. However, in real world, humans apply different actions to different parts of objects. E.g. to get some coffee from a coffee machine, a human may first press the power button to open the machine, and press the brew button afterwards to brew coffee. Thus we design the objects in our system in a compositional way, i.e., an object has multiple components, each of which has its own affordance. This extends the typical action space in prior systems to a much larger set of fine-grained actions and enables the agents to learn object-related causality and commonsense.

### 3.3. User Interface

With a detailed agent embodiment representation, multiple levels of human-object-interactions are available. In particular, there are two ways for users and algorithms to control the agent:

(1) Users and algorithms can directly control the agent's head and hands. During teleoperation, actions are recorded using a set of off-the-shelf VR device, in our case, an Oculus Rift head-mounted display (HMD) and a pair of Oculus Touch controllers. Two Oculus constellation sensors are used to track the transforms of the headset and controllers in 3D spaces. We then apply the data to a human avatar in the virtual environment: the avatar's head and hand movements correspond to the human user's, while other parts of its body are animated through a built-in Inverse Kinematics solver (Forward And Backward Reaching Inverse Kinematics, or FABRIK). Human users are free to navigate the space using the Thumbsticks and grab objects using the Trigger button on the controller. Figure 6 gives an example of collecting demonstrations for continuous actions.

(2) The Python API offers a way to control the agent by sending discrete action sequences. In particular, it provides world states and receives discrete action sequences. The

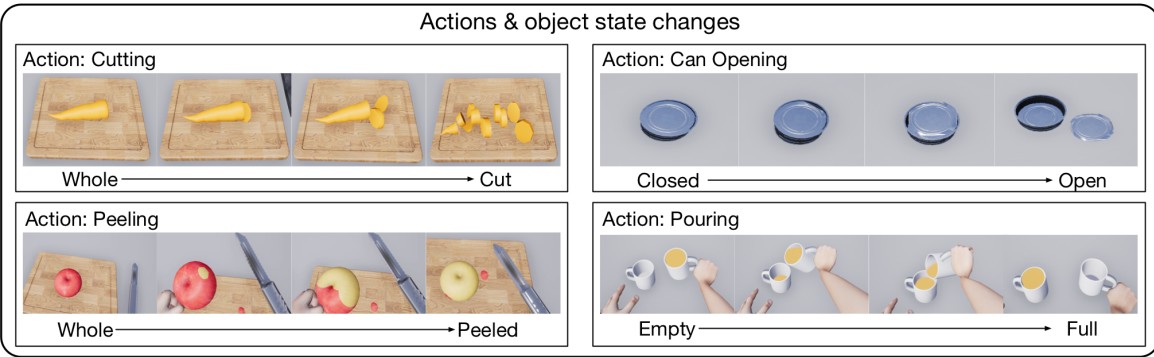

*Figure 5.* Sample actions and object state changes by making use of different tools in VRKitchen.

world state is comprised of the locations and current states of nearby objects and a RGB/depth image of agent's first person view. Figure 7 shows examples of recorded human demonstrations for tasks *pizza* from a third person view.

### 3.4. Python-UE4 Bridge

The Python-UE4 bridge contains a communication module and a controller. The Python server communicates with the game engine to receive data from the environment and send requests to the agent. It is connected to the engine through sockets. To perform an action, the server sends a command to UE4 and waits for response. A client in UE4 parses the command and applies the corresponding animations to the agent. A payload containing states of nearby objects, agent's first person camera view (in terms of RGB, depth and object instance segmentations) and other task-relevant information are sent back to the Python server. The process repeats until terminal state is reached.

The controller enables both low level motor controls and high level commands. Low level controls change local translation and rotation of agent's body, heads and hands, while other body parts are animated using FABRIK. High level commands, which performs discrete actions such as taking or placing an object, are further implemented by taking advantage of the low level controller. To cut a carrot with a knife, for example, the high level controller iteratively updates the hand location until the knife reaches the carrot.

### 3.5. Performance

We run VRKitchen on a computer with Intel(R) Core(TM) i7-7700K processor @ 4.50GHz and NVIDIA Titan X (Pascal) graphics card. A typical interaction, including sending command, executing the action, rendering frame and getting response, takes about 0.066 seconds (15 actions per second) for a single thread. The resolutions for RGB, depth and object segmentation images are by default 84×84.

## 4. VR Chef Challenge

In this paper, we propose the VR chef challenge consisting of two sets of cooking tasks: (a) tool use, where learning motor control is the main challenge; and (b) preparing dishes, where compositional goals are involved and there are hidden task dependencies (e.g., ingredients need to be prepared in a certain order). The first set of tasks requires an agent to continuously control its hands to make use of a tool. In the second set of tasks, agents must perform a series of atomic actions in the right order to achieve the final goal.

### 4.1. Tool Use

Based on available actions and state changes in the environment (shown in Figure 5), we have designed 5 tool use tasks: *cutting, peeling, can-opening, pouring* and *getting water*. These tasks are common in cooking and require accurate control of agent's hand to change the state of an object. Agents would get rewards once it takes the correct tool and each time states of objects being changed. Definitions for these task are displayed as following.

**Cutting**: cut a carrot into four pieces with a knife. The agent gets reward from getting the knife and each cutting.

**Peeling**: peel a kiwi with a peeler. The agent receives reward from getting the peeler and each peeled skin. Note that the skin will be peeled only if the peeler touches it within a certain range of rotation. The task finishes if enough pieces of skins are peeled.

**Can-opening**: open a can with a can opener. Around the lid, there are four sides. One side of the lid will break if it overlaps with the blade. Agents receive reward from taking the opener and breaking each side of the lid.

**Pouring**: take a cup full of water and pour water into a empty cup. The agent is rewarded for taking the full cup and each additional amount of water added into the empty cup. The task is considered done only if the cup is filled over fifty percent.

**Getting water**: take an empty cup and get water from a running tap. The agent is rewarded for taking the cup and

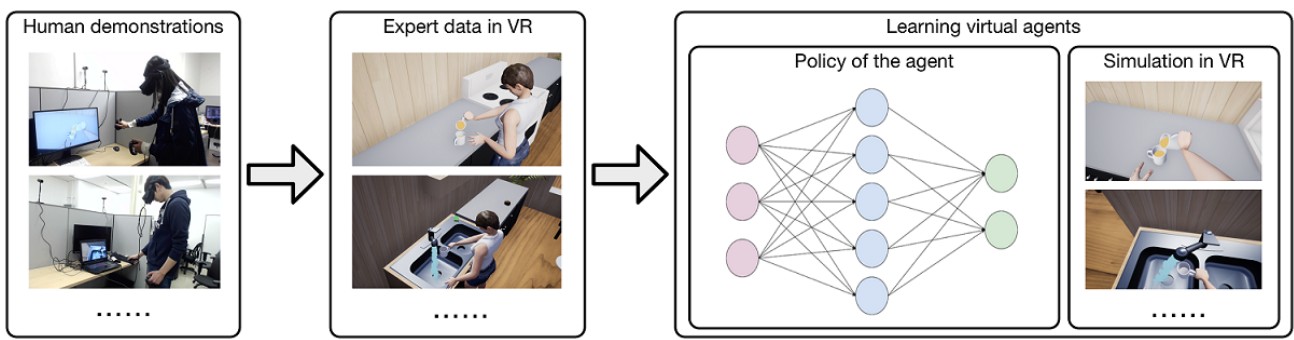

*Figure 6.* Users can provide demonstrations by doing tasks in VRKitchen. These data can be taken to initialize the agent's policy, which will be improved through interactions with the virtual environment.

each additional amount of water added into it. The task is considered done only if the cup is filled over fifty percent.

In each episode, the agent can control the translation and rotation of avatar's right hand for 50 steps. The continuous action space is defined as a tuple $(\Delta x, \Delta y, \Delta z, \Delta\phi, \Delta\theta, \Delta\psi, \gamma)$, where $(x, y, z)$ is the right hand 3D location and $(\phi, \theta, \psi)$ is the 3D rotation in terms of Euler angle. If the grab strength $\gamma$ is bigger than a threshold (0.1 in our case), objects within a certain range of avatar's hand will be attached to a socket. Physics simulations are enabled on all the objects. For objects attached to agent's hand, physics simulation is disabled.

### 4.2. Preparing Dishes

Applying reinforcement learning to complex real life tasks require agents to take advantage of a sequence of atomic actions to reach a certain goal. Many challenges arise in this domain, including making long explorations and visual understanding of the surroundings. In VRKitchen, we design all atomic actions and object state changes available in several dish preparing tasks. Using these atomic actions, the agent can interact with the environments until a predefined goal is reached. Figure 8 shows some examples of dishes.

#### 4.2.1. ATOMIC ACTIONS

Each atomic action listed below can be viewed as a composition of a verb (action) and a noun (object). Objects can be grouped into three types: *tools, ingredients and receptacles*. (1) *Ingredients* are small objects needed to make a certain dish. We assume that the agent can hold at most one *ingredient* at a time. (2) For *receptacles*, we follow the definition in (Kolve et al., 2017). They are defined as stationary objects which can hold things. Certain *receptacles* are called *containers* which can be closed and agents can not interact with the objects within them until they are open. (3) *Tools* can be used to change the states of certain *ingredients*. Atomic actions and object affordance are defined in a following way:

`Take` {*ingredient*}: take an *ingredient* from a nearby *recep-*

*tacle*;

`Put into` {*receptacle*}: put a held *ingredient* into a nearby *receptacle*;

`Use` {*tool*}: use a *tool* to change the state of a *ingredient* in a nearby *receptacle*;

`Navigate` {*tool, receptacle*}: move to a *tool* or *receptacle*;

`Toggle` {*container*}: change state of a *container* in front of the agent.

`Turn`: rotating the agent's facing direction by 90 degrees.

Note that actions including `Take`, `put into`, `use`, and `toggle` would fail if the agent is not near the target object.

#### 4.2.2. *Ingredient* SETS AND STATES

Meanwhile, there are seven sets of *ingredients*, including *fruit, meat, vegetable, cold-cut, cheese, sauce, bread* and *dough*. Each set contains a number of *ingredients* as variants: for example, *cold-cut* can be ham, turkey or salami. One *ingredient* may have up to four types of state changes: *cut, peeled, cooked* and *juiced*. We manually define affordance for each set of *ingredients*: e.g. *fruit* and *vegetable* like oranges and tomatoes can be juiced (using a juicer) while bread and meat can not. *Tools* include grater, juicer, knife, oven, sauce-bottle, stove and *receptacles* are fridge, plate, cut-board, pot and cup.

#### 4.2.3. DISHES

Based on the atomic actions defined in 4.2.1, agents can prepare five dishes: *fruit juice, stew, roast meat, sandwich* and *pizza*. Goals of each tasks are compositionally defined upon (1) goals states of several sets of ingredients and (2) target locations: to fulfill a task, all required *ingredients* should meet the goal states and be placed in a target location. For example, to fulfill the task *fruit juice*, two *fruits* should be cut, juiced and `put into` the same cup. Here, the target locations are one or several kinds of *containers*.

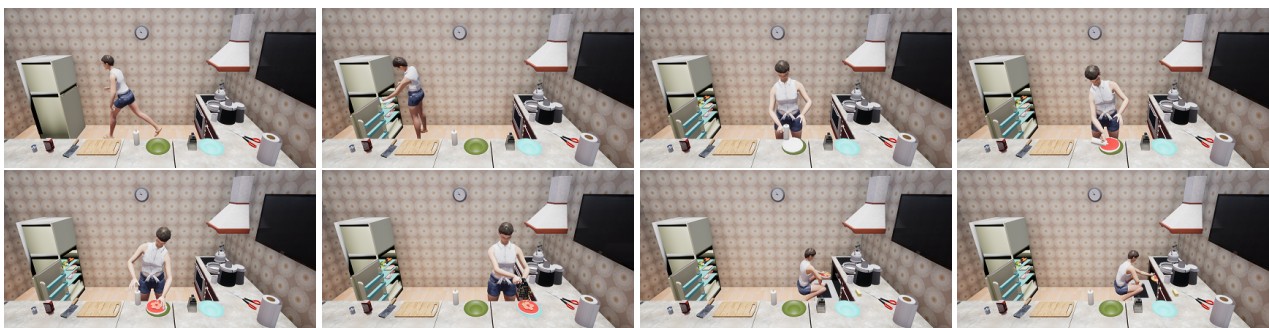

*Figure 7.* An example of human demonstrations for making a *pizza*, which is one of the dishes.

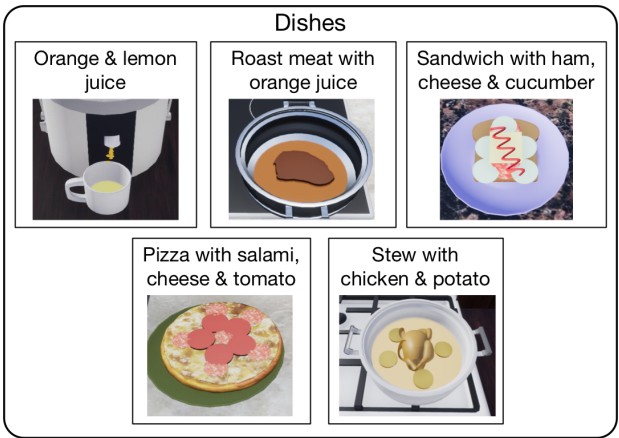

*Figure 8.* Examples of dishes made in VRKitchen. Note that different *ingredients* leads to different variants of a dish. For example, mixing orange and kiwi juice together would make *orange & kiwi juice*.

## 5. Benchmarking VR Chef Challenge

We train agents in our environments using several popular deep reinforcement learning algorithms to provide benchmarks of proposed tasks.

### 5.1. Experiment 1: Using Tools

#### 5.1.1. EXPERIMENT SETUP

In this experiment, we are learning motor controls for an agent to use different tools. In particular, five tasks (defined in 4.1) are available, including (a) cutting a carrot; (b) peeling a kiwi; (c) opening a can; (d) pouring water from one cup to another; (e) getting water from the tap. Successful policies should first learn to take the *tool* and then perform a set of transformations and rotations on the hand, correspond to the task and surroundings.

#### 5.1.2. RESULTS AND ANALYSIS

For five tool use tasks, we conduct experiments using three deep reinforcement learning algorithms: A2C (Mnih et al., 2016), DDPG (Lillicrap et al., 2015), PPO (Schulman et al., 2017). The inputs are the $84 \times 84$ raw pixels coming

from agent's first person view. We run each algorithm for 10000 episodes, each of which terminates if the goal state is reached or the episode exceeds 1000 steps.

Figure 9 summarizes the results of our experiments. We see that because of the large state space, agents trained using RL algorithms rarely succeed in most of the five tasks.

### 5.2. Experiment 2: Preparing Dishes

#### 5.2.1. EXPERIMENT SETUP

In this experiment, we study visual planning tasks, which require the agent to emit a sequence of atomic actions to meet several sub-goals. In general, successful plans should first go to locations near some *ingredients*, take them and change their states by making use of some *tools*. Particularly, tasks have three levels of difficulty:

1. Easy: First, Navigate to a *receptacle* $R_1$ and take an *ingredient* $I_1$. After that, Navigate to a *tool* $T_1$ with $I_1$ and use $T_1$. An example would be making orange juice: the agent should first go to the fridge and take an orange. Then it should take the orange to the juicer and use it. This task requires the agent to reason about the causal effects of its actions.
2. Medium: In addition to the "Easy" task, this task requires the agent to take from the *receptacle* $R_1$ a different *ingredient* $I_2$. The task ends when the agent puts $I_1$ and $I_2$ into a new receptacle $R_2$. A sample task is making beef stew: the agent should first go to the fridge and take an tomato and beef. Then it should bring the tomato to the knife and use it. Finally, the agent should put both beef and tomato into a pot. This task requires identifying various tools, receptacles and ingredients.
3. Hard: Compared to the "Medium" tasks, more objects are involved in hard tasks. Moreover, a longer sequence of actions is required to reach the goal state. Making sandwich is one example: *ingredients* involved are bread, tomato, ham and cheese, and an optimal policy takes about 29 steps to reach the goal states.

#### 5.2.2. RESULTS AND ANALYSIS

We evaluate the performance of three deep reinforcement learning algorithms (A2C (Mnih et al., 2016), DQN (Mnih

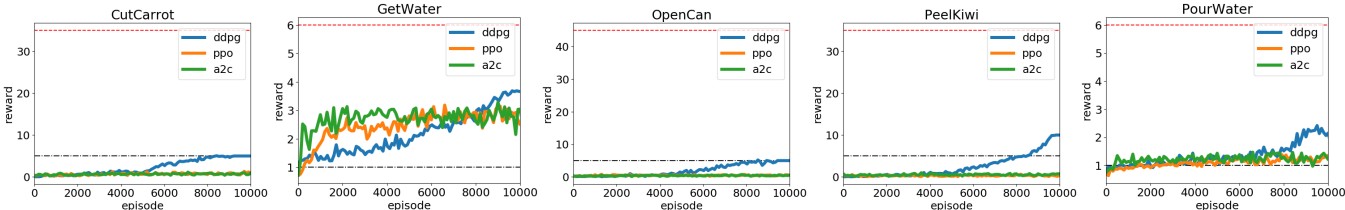

*Figure 9.* Experiment results for five tool use tasks. Black horizontal lines show rewards agents get from taking the tools, and the red lines indicate the rewards of completing the whole tasks. Each curve shows the average reward an agent receives using one of three different RL algorithms.

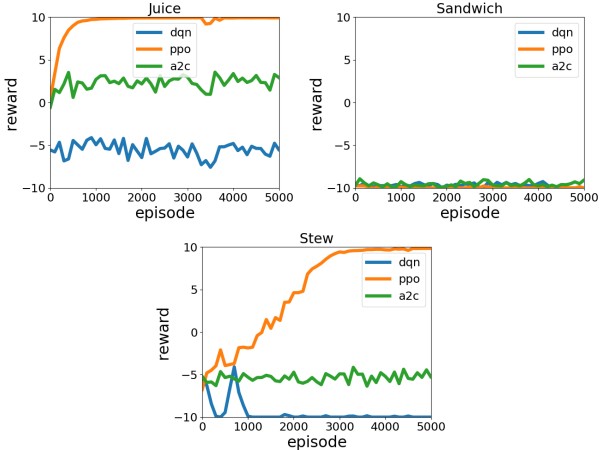

*Figure 10.* Experiment results for three dish preparing tasks. Each curve shows the average reward an agent receives using one of RL algorithms.

et al., 2015) and PPO (Schulman et al., 2017)) on dish preparing tasks. We run each algorithm for 5000 episodes. We consider an episode fails if it exceeds 1000 steps.

Figure 10 shows the experiment results. For easy tasks (*juice*), it takes less than 1000 episodes for the best algorithm to find near-optimal solution. For medium-level tasks (*stew*), PPO (Schulman et al., 2017) is still able to converge after 3000 episodes. None of three RL algorithms can successfully guide the agent in hard tasks.

## 6. Human Demonstration Collection

We compiled a human demonstration dataset for both cooking tasks – UCLA VR Chef Dataset. We took advantage of the user interface to collect these demonstrations (by VR device and Python API, described in 3.3). We leave learning from these demonstrations to future work.

For tool use, we collected 20 human demonstrations for each tasks. Most users involved had little or no experience using VR device. Prior to the collection, users were provided with instructions on how to put on the HMD and how to interact with objects using the Touch controller. For each task, they were given 5 minutes to get familiar with it before demonstrations were collected. The program recorded 3D

locations, rotations and grab strength for avatar's hands every 0.01 seconds. Actions were computed using the relative value between two time stamps, while states come from user's first person camera view. Examples of users doing tasks in the virtual environment can be found in the left part of Figure 6.

To collect demonstrations for preparing dishes, each participant is first assigned a scene and a task, then watches a demonstration video containing a successful trial. Afterwards, users are asked to interact with the scenes and complete the tasks. For visualization purposes, we design a web-based user interface which utilizes the Python API to collect demonstrations. The web-based UI displays the world states (including the agent's egocentric view and the states of nearby objects) to the user. Users can then perform discrete action sequences which would be transferred into motor control signals and sent to the agents through the server. We allow users to freely take advantage of all the *tools* and *ingredients* available in the scenes, thus there may be multiple ways to prepare a certain dish. Atomic action sequences are recorded for every legal moves. There are 20 demonstrations for each of the five dishes. On average, each demonstration has 25 steps. Figure 7 shows a sample sequence of recorded demonstrations.

## 7. Conclusion

We have designed a virtual reality system, VRKitchen, which offers physical simulation, photo-realistic rendering of multiple kitchen environments, a large set of fine-grained object manipulations, and embodied agents with human-like appearances. We have implemented toolkits for training and testing AI agents as well as for collecting human demonstrations in our system. By utilizing our system, we have proposed VR chef challenge with two sets of real life cooking tasks, on which we benchmarked the performance of several popular deep reinforcement learning approaches. We are also able to compile a video dataset of human demonstrations of the cooking tasks using the user interface in the system. In the future, we plan to enrich the simulation in our system and conduct a more thorough evaluation of current reinforcement learning and imitation learning approaches.

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
