# OpenReview forum: "VRKitchen: an Interactive 3D Environment for Learning Real Life Cooking Tasks"
_ICML.cc/2019/Workshop/RL4RealLife — RL4RealLife 2019_

### Official Review · AnonReviewer1 · 2019-05-23
**The author trained agent to play cooking game in VR system. By utilizing VR system for its environment, the agent can be exposed to dynamic real life environment and human teachers can provide demonstrations for training agents.**

**Rating:** 5
**Confidence:** 4

**Review:**

Pros:
- This paper suggested novel approach to utilize VR to train agents which has several advantages - (1) agent can learn visual representation of a dynamic environment (more realistic), (2) agent can learn to generate long-term plans for complex task, (3) agent can learn from human demonstrations to bootstrap agent's models. Those advantages are very clear, convincing and novel. Especially, the idea using human demonstration to train agent by combining imitation learning is fascinating.

- The paper is clearly written with good visual representations and contributions are strong.
- The author suggested to use three different RL methods, including A2C, DDPG, PPO and compare each other.

Cons:
- Compared with the novel approaches and strong contributions, the experimental results are not enough. It would be great if we can compare results between with and without human demonstrations.
-Figure 9 and Figure 10 is too small and difficult to see.

Overall, very nice work and I would like to give strong accept!

---

### Official Review · AnonReviewer2 · 2019-05-25
**In this project, the authors design and implement a virtual reality kitchen system with physics engines, user interface, benchmarks, and two challenges.**

**Rating:** 5
**Confidence:** 3

**Review:**



In the VR kitchen project, the authors design and implement a virtual reality kitchen system that integrates functions for reinforcement leaning agents to conduct a series of complex cooking tasks and human teachers to demonstrate cooking tasks for training. In comparison to related environments, this project provides an environment that follows physics principles, realistic rendering, state changes, object manipulations, avatar, demo capability, and benchmarks of cooking tasks using three widely used deep reinforcement learning algorithms: A2C, DDPG, and PPO. The authors present two cooking tasks, including using tools and preparing dishes.

The work is valuable to the community that allows additional participants to more easily conduct cooking robot related reinforcement learning research. Main functionalities are provided in the environment. I consider the work a useful contribution to the workshop.

---

### Decision · Program_Chairs · 2019-05-28

Accept